# Pre-gestational diabetes: Maternal body mass index and gestational weight gain are associated with augmented umbilical venous flow, fetal liver perfusion, and thus birthweight

**Agnethe Lund**[1,2], **Cathrine Ebbing**[1,2]*, **Svein Rasmussen**[1,2], **Elisabeth Qvigstad**[3,4], **Torvid Kiserud**[1,2], **Jörg Kessler**[1,2]

1 Department of Obstetrics and Gynecology, Haukeland University Hospital, Bergen, Norway, 2 Department of Clinical Science, Research Group for Pregnancy, Fetal Development and Birth, University of Bergen, Bergen, Norway, 3 Department of Endocrinology, Morbid Obesity and Preventive Medicine, Oslo University Hospital, Oslo, Norway, 4 Faculty of Medicine, Institute of Clinical Medicine, University of Oslo, Oslo, Norway

* Cathrine.Ebbing@helse-bergen.no

**Data Availability Statement:** The combination of detailed clinical information in our study could

## Abstract

### Objectives

To assess how maternal body mass index and gestational weight gain are related to on fetal venous liver flow and birthweight in pregnancies with pre-gestational diabetes mellitus.

### Methods

In a longitudinal observational study, 49 women with pre-gestational diabetes mellitus were included for monthly assessments (gestational weeks 24–36). According to the Institute Of Medicine criteria, body mass index was categorized to underweight, normal, overweight, and obese, while gestational weight gain was classified as insufficient, appropriate or excessive. Fetal size, portal flow, umbilical venous flow and distribution to the fetal liver or ductus venosus were determined using ultrasound techniques. The impact of fetal venous liver perfusion on birthweight and how body mass index and gestational weight gain modified this effect, was compared with a reference population (n = 160).

### Results

The positive association between umbilical flow to liver and birthweight was more pronounced in pregnancies with pre-gestational diabetes mellitus than in the reference population. Overweight and excessive gestational weight gain were associated with higher birthweights in women with pre-gestational diabetes mellitus, but not in the reference population. Fetuses of overweight women with pre-gestational diabetes mellitus had higher umbilical ($p$ = 0.02) and total venous liver flows ($p$ = 0.02), and a lower portal flow fraction ($p$ = 0.04) than in the reference population. In pre-gestational diabetes mellitus pregnancies

enable identification of specific participants. Therefore, data sharing must be approved by our ethics committee, even if the data are de-identified. The Regional Committee for Medical and Health Research Ethics (REK Vest) can be contacted referring to the number REK vest 2011/2030; post@helseforskning.etikkom.no. The rules and procedures can be found here: https://helseforskning.etikkom.no/reglerogrutiner/loverogregler?p_dim=34770&_ikbLanguageCode=us.

**Funding:** This research was financed by the Western Norway Regional Health Authority, Helse Vest, project number 911765, https://helse-vest.no/vart-oppdrag/vare-hovudoppgaver/forsking/forskingsmidlar. This funded the PhD work for A.L., main author of the submitted paper. The funders had no role in study design, data collection and analysis, decision to publish, or preparation of the manuscript.

**Competing interests:** The authors have declared that no competing interests exist.

**Abbreviations:** BMI, Body Mass Index; GWG, Gestational Weight Gain; IOM, Institute Of Medicine; PGDM, Pre-gestational Diabetes Mellitus.

with excessive gestational weight gain, the umbilical flow to liver was higher than in those with appropriate weight gain ($p = 0.02$).

## Conclusions

The results support the hypothesis that umbilical flow to the fetal liver is a key determinant for fetal growth and birthweight modifiable by maternal factors. Maternal pre-gestational diabetes mellitus seems to augment this influence as shown with body mass index and gestational weight gain.

## Introduction

In pregnancies with pre-gestational diabetes mellitus (PGDM), the risk of adverse perinatal outcome is increased [1], and complications are often associated with large for gestational age neonates [2,3]. Since hyperglycemia may cause accelerated fetal growth, optimal glycemic control is a cornerstone in the clinical follow-up [4,5]. However, in PGDM populations with apparently good glycemic control the incidence of large neonates remains high [6]. Recent improvements in glucose monitoring demonstrate that reduced glucose excursions/variability improve pregnancy outcomes [5].

These women have on average higher pre-pregnancy body mass index (BMI) and more gestational weight gain than women without diabetes mellitus [7,8]. Overweight and obesity add significantly to the risk of large for gestational age offspring in these pregnancies [7], and excess gestational weight gain is linked to risk for neonatal macrosomia independent of glycemic control in women with type 1 diabetes [8]. Thus, women with PGDM are advised to aim for pre-pregnancy BMI in the normal range, less gestational weight gain than women without diabetes, and strict glycemic control [5,8,9].

A known mechanism regulating fetal growth is the distribution of umbilical venous blood to the fetal liver (Fig 1) [10,11]. This blood, high in nutrition and oxygen, is directed either to the fetal liver or shunted through the ductus venosus supplying the fetal heart and brain (Fig 1). In low-risk pregnancies, at average 70–80% of the umbilical venous return is distributed to the liver [12–14]. Experimentally increased umbilical flow to the fetal liver induces hepatic cell proliferation and production of IGF-1 and -2 that is followed by augmented growth of heart, skeletal muscle and kidneys [15]. In humans, a higher umbilical flow to the liver is associated with newborn adiposity [16]. The distribution of the umbilical blood is influenced by maternal BMI in pregnancies without diabetes [17]. In normal weight women the maternal-fetal glucose gradient was found to correlate with the distribution of the umbilical flow to the fetal liver, while in overweight mothers no such correlation was found [18]. In pregnancies with PGDM, we found that the proportion of umbilical venous return distributed to the fetal liver was graded according to maternal $HbA_{1C}$ [19]. However, whether maternal BMI and gestational weight gain in women with PGDM influence this distributional mechanism is not known.

The aim of the present study was to assess the relation between fetal venous liver flow and birthweight in PGDM pregnancies, and how this relation is modified by BMI and gestational weight gain.

## Materials and methods

The present prospective longitudinal observational study was part of the project DiaDoppler investigating fetal hemodynamics in pregnancies with PGDM. We have previously reported

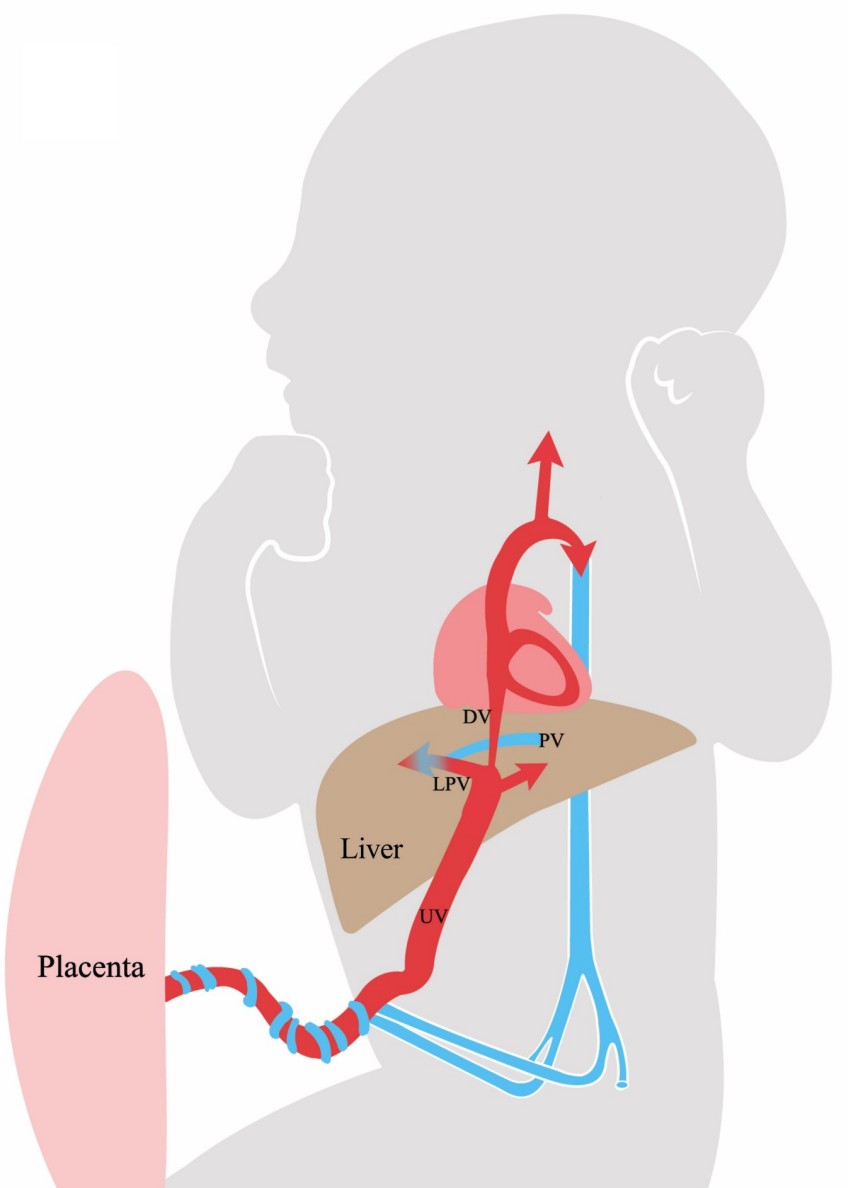

**Fig 1. The fetal umbilical venous circulation schematic.** Well-oxygenated and nutrient rich blood (red) from the placenta reaches the fetus through the umbilical vein (UV). This blood is distributed either to the fetal liver (arrows within the liver) or shunted through the ductus venosus (DV) to supply the heart and brain. The portal vein (PV) carries low-oxygenated blood (blue) from the visceral organs and blends in with the umbilical blood from the left portal branch (LPV) to supply the right liver lobe.

the development of the ductus venosus, umbilical and portal blood flows during the second half of pregnancy in this population [19,20]. Here we assess whether maternal BMI and gestational weight gain are associated with modification of the venous perfusion of the fetal liver and birthweight.

## Subjects

In our region, all pregnant women with PGDM are referred to the tertiary center at Haukeland University Hospital for follow-up by a multidisciplinary team. All women with PGDM and

singleton pregnancies during the period August 2013 to May 2016 were invited to participate in the study. The study protocol was approved by the Regional Committee for Medical Research Ethics (REK vest 2011/2030), and 52 women (74% of those invited) gave informed written consent. All participants used insulin treatment during pregnancy. Forty-four participants had type 1 and eight had type 2 diabetes. Three participants with type 2 diabetes withdrew after the first visit, thus 49 women with PGDM constituted our study population.

Information on maternal height and pre-pregnancy weight was self-reported and collected from medical records. Pre-pregnancy BMI (weight (kg)/height $(m)^2$) was categorized according to the Institute Of Medicine (IOM) guidelines: underweight ($<18.5$), normal weight (18.5–24.9), overweight (25–29.9) and obese ($\geq30$) [21]. Weekly gestational weight gain was calculated by subtracting pre-pregnancy weight from the last weight measured before delivery, divided by gestational age at the last weighing. Weekly gestational weight gain was categorized according to pre-pregnancy BMI and the IOM guideline as insufficient, appropriate or excessive [21].

Gestational age was determined by measuring the crown rump length [22], using a vaginal ultrasound probe (Vivid 7, GE Healthcare Vingmed Ultrasound, E8C, 8 MHz) around week 9 of pregnancy. $HbA_{1C}$ was measured at inclusion in the first trimester. Birthweight z-scores were calculated according to gestational age at delivery [23]. Information on maternal $HbA_{1C}$, birthweight, neonatal acidosis at birth, mode of delivery, Apgar score and transfer to the neonatal ward was collected from clinical records.

**Flow variables.**  The ultrasound and Doppler examinations were performed at gestational weeks 24, 28, 32 and 36. Using an abdominal transducer (Vivid 7, GE Healthcare Vingmed Ultrasound, Horten, Norway) (M4S, 2.0–4.3 MHz), the fetal vein diameters and blood flow velocities were measured to calculate the blood flow volumes. Measurement techniques and formulas used for the calculations are reported previously [24,25].

**Statistics.**  BMI, weekly maternal weight gain and fetal flows in the study population were compared with reference ranges (obtained in a longitudinal study of 160 healthy pregnancies using identical methods by our research group) [14,24,25]. We tested whether $HbA_{1C}$ differed between the BMI and gestational weight gain groups.

Multilevel regression was used to calculate the main outcome fetal blood flow by gestational age [14,23]. We used log-likelihood test to assess whether adding BMI or gestational weight gain categories significantly influenced the longitudinal development of flow by gestational age. Since only two participants with PGDM were underweight, this group was excluded from the log-likelihood analyses. Flow variable categories (tertiles) were defined by the distribution in the low-risk reference population. Differences in birthweight between flow tertiles, and between BMI and weekly gestational weight gain categories, were estimated using analysis of variance. Relations between birthweight z-scores and the exposures, BMI and gestational weight gain were assessed as continuous variables in regression analyses.

The statistical analyses were performed with the Statistical Package for the Social Sciences (version 24, SPSS, Chicago, IL) and the MLWin program (version 2.35, Centre of Multilevel Modeling, University of Bristol, UK). P-values $<0.05$ were considered significant.

## Results

Characteristics of the study and reference populations at inclusion are shown in Table 1 and have been described previously [14,19]. The birthweight z-score distributions by BMI and gestational weight gain categories are presented in Table 2.

At inclusion median $HbA_{1C}$ was 6.70% (50 mmol/L) (range 4.90–12% (30–108 mmol/L)) and median duration of diabetes 17 years (range 1–37 years). The mean difference between

**Table 1. Maternal and neonatal characteristics and outcomes in the study population of 49 pregnancies with pregestational diabetes mellitus.**

|  | Number | Percent |
|---|---|---|
| Type 1 diabetes mellitus | 44 | 89.8 |
| Type 2 diabetes mellitus | 5 | 10.2 |
| Maternal diabetic complications | 9 | 18.4 |
| Hypothyroidism | 9 | 18.4 |
| Chronic hypertension | 7 | 14.3 |
| Preeclampsia | 3 | 6.1 |
| Preterm birth* | 15 | 30.6 |
| Cesarean section | 22 | 44.9 |
| Metabolic acidosis at birth † | 1 | 2 |
| 5-min Apgar score <7 | 1 | 2 |
| Transfer to neonatal intensive care ward | 20 | 40.8 |
| Perinatal death ‡ | 1 | 2 |
| Malformation § | 2 | 4 |

*Preterm birth, gestational age <37 weeks

† Metabolic acidosis defined as an umbilical arterial pH of <7.0 and a base deficit of >12.

‡Intrauterine fetal death at gestational week 36.

§One neonate with sagittal craniosynostosis and one with congenital heart defect (anomalous left coronary artery from the pulmonary artery).

measured weight at inclusion (at median gestational age 9.4 weeks) and the self-reported prepregnancy weight in the study population was 2.0 kg. There was no difference in $HbA_{1C}$ between the various BMI or gestational weight gain categories, $p = 0.72$ and $p = 0.35$ respectively. The gestational age at birth was lower in the study population than in the reference population, 37.8 weeks and 40.3, respectively [14].

**Table 2. Distribution of BMI and GWG categories and birthweight z-scores in the healthy reference and the PGDM populations.**

|  |  | Reference Median (range) |  |  | PGDM Median (range) |  |  |
|---|---|---|---|---|---|---|---|
| BMI ($kg/m^2$) |  | 23.0 (17.0–41.0) |  |  | 25.4 (19.8–44.1) |  |  |
| GWG/week (kg/week) |  | 0.37 (0.01–0.73) |  |  | 0.46 (-0.14–0.95) |  |  |
|  | Category |  | % | Mean BW z-score | n | % | Mean BW z-score |
| BMI | normal weight | 101 | 63.1 | -0.11 | 22 | 44.9 | 0.62 |
|  | overweight | 43 | 26.9 | 0.17 | 14 | 28.6 | 2.02 |
|  | obese | 9 | 5.6 | -0.52 | 11 | 22.4 | 0.59 |
| $p^*$ |  |  |  | 0.224 |  |  | 0.001* |
| GWG | insufficient | 47 | 29.4 | -0.16 | 6 | 12.2 | 0.31 |
|  | appropriate | 61 | 38.1 | -0.08 | 16 | 32.7 | 0.60 |
|  | excessive | 47 | 29.4 | 0.10 | 27 | 55.1 | 1.48 |
| $p^*$ |  |  |  | 0.556 |  |  | 0.008* |
| Total group |  | 155 |  | -0.06 (-3.02–1.81) | 49 |  | 1.05 (-2.15–5.82) |

PGDM, pregestational diabetes; Body Mass Index, BMI; BMI categories were defined as: normal weight (18.5–25), overweight (25–30), obese (≥30); Gestational Weight Gain, GWG; GWG categories were defined as: insufficient, appropriate, excessive

* Mean birthweight z-score difference between categories tested by univariate linear regression (one-way ANOVA)

### Fetal venous flow and birthweight

In both the reference and PGDM populations, fetal venous liver flow was positively related to birthweight, but the association to birthweight was more pronounced in pregnancies with PGDM (Fig 2 and Table 3).

### BMI, gestational weight gain and birthweight

In women with PGDM, overweight and excessive weight gain were associated with higher birthweight, which was not evident in the reference population (Table 4). In the PGDM population, 39% of the neonates had developed macrosomia (birthweight >90th percentile), and 8% were small for gestational age (<10th percentile), compared with 7 and 14%, respectively, in the reference population [23].

In PGDM, the relation between BMI and birthweight had an inverted U-shape, with the highest mean birthweight z-score in the overweight group (Fig 3). Within the PGDM population, neonates of obese women weighed less than those in the overweight group. Still, these

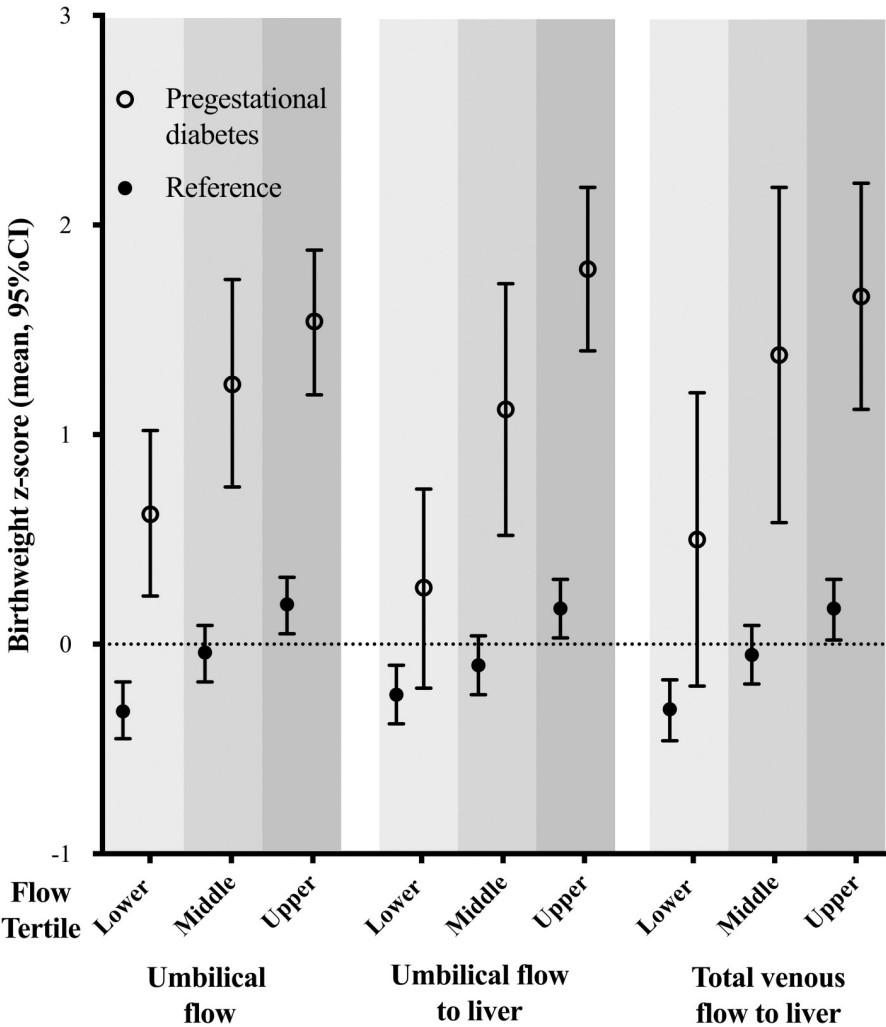

**Fig 2. Birthweight z-scores in fetal flow tertiles in the study population with pregestational diabetes mellitus (PGDM) and the reference group.** Flow variables were divided into tertiles defined by the distribution in the reference group.

**Table 3. Birthweight z-scores according to fetal flow tertiles in the reference and the pregestational diabetes mellitus population (160 and 49 participants, respectively).**

| | Flow tertiles | Birthweight z-scores | | | | | | p-value[†] |
|---|---|---|---|---|---|---|---|---|
| | | Reference | | | Pregestational diabetes | | | |
| | | N | mean | CI | n | mean | CI | |
| Umbilical flow | lower | 191 | -0.32 | -0.45 – -0.18 | 66 | 0.62 | 0.23–1.02 | <0.0001 |
| | middle | 192 | -0.04 | -0.18–0.09 | 41 | 1.24 | 0.75–1.74 | <0.0001 |
| | upper | 191 | 0.19 | 0.05–0.32 | 85 | 1.54 | 1.19–1.88 | <0.0001 |
| p* | | | <0.001 | | | 0.003 | | |
| Umbilical flow to liver | lower | 185 | -0.24 | -0.38 – -0.10 | 40 | 0.27 | -0.21–0.74 | 0.007 |
| | middle | 185 | -0.10 | -0.24–0.04 | 25 | 1.12 | 0.52–1.72 | <0.0001 |
| | upper | 185 | 0.17 | 0.03–0.31 | 58 | 1.79 | 1.40–2.18 | <0.0001 |
| p* | | | <0.001 | | | <0.001 | | |
| Ductus venosus flow | lower | 181 | 0.18 | 0.04–0.32 | 62 | 1.28 | 0.86–1.70 | <0.0001 |
| | middle | 181 | -0.12 | -0.26–0.23 | 25 | 0.85 | 0.19–1.51 | <0.0001 |
| | upper | 181 | -0.25 | -0.39 – -0.11 | 51 | 1.14 | 0.68–1.61 | <0.0001 |
| p* | | | <0.001 | | | 0.548 | | |
| Ductus venosus fraction | lower | 178 | -0.07 | -0.22–0.07 | 62 | 1.47 | 1.06–1.88 | <0.0001 |
| | middle | 178 | -0.01 | -0.15–0.14 | 28 | 1.05 | 0.44–1.66 | <0.0001 |
| | upper | 178 | -0.09 | -0.24–0.05 | 33 | 0.66 | 0.10–1.22 | 0.001 |
| p* | | | 0.671 | | | 0.067 | | |
| Left portal vein blood velocity[¥] | lower | 184 | -0.26 | -0.40 - -0.12 | 38 | 0.62 | 0.10–1.13 | <0.0001 |
| | middle | 185 | 0.03 | -0.12–0.17 | 51 | 0.84 | 0.39–1.28 | <0.0001 |
| | upper | 184 | 0.07 | -0.07–0.22 | 113 | 1.44 | 1.14–1.74 | <0.0001 |
| p* | | | 0.002 | | | 0.009 | | |
| Portal vein flow | lower | 186 | -0.41 | -0.55 - -0.27 | 35 | 1.45 | 0.89–2.00 | <0.0001 |
| | middle | 186 | -0.01 | -0.14–0.14 | 19 | 0.77 | 0.02–1.53 | 0.003 |
| | upper | 186 | 0.20 | 0.07–0.34 | 40 | 1.26 | 0.74–1.78 | <0.0001 |
| p* | | | <0.001 | | | 0.364 | | |
| Portal vein fraction | lower | 174 | -0.12 | -0.26–0.03 | 34 | 1.74 | 1.17–2.31 | <0.0001 |
| | middle | 173 | -0.05 | -0.20–0.10 | 9 | 0.73 | -0.38–1.83 | 0.021 |
| | upper | 173 | -0.05 | -0.20–0.10 | 33 | 0.91 | 0.33–1.49 | <0.0001 |
| p* | | | 0.761 | | | 0.085 | | |
| Total venous flow to liver | lower | 175 | -0.31 | -0.46 - -0.17 | 22 | 0.499 | -0.20–1.20 | 0.001 |
| | middle | 175 | -0.05 | -0.19–0.09 | 17 | 1.380 | 0.58–2.18 | <0.0001 |
| | upper | 175 | 0.17 | 0.02–0.31 | 37 | 1.656 | 1.12–2.20 | <0.0001 |
| p* | | | <0.001 | | | 0.037 | | |

Flow variables were divided into tertiles defined by the distribution in the reference population (upper, middle, lower), n; total number of observations

*Birthweight z-score difference between fetal blood flow tertiles tested by ANOVA within each population (table read vertically)

[†] Birthweight z-score difference between the reference and study populations in flow tertiles tested by independent sample T-test (table read horizontally); CI,

confidence interval; Flow, volume blood flow (mL/min); z-score, standard deviation score

[¥] Flow velocity, time-averaged maximum blood velocity (cm/sec).

neonates had a larger birthweight z-score than the obese of the reference group (mean z-scores difference 1.11, $p = 0.045$) (Table 1).

In the PGDM population, there was a positive linear relation between weekly gestational weight gain and z-scores of birthweights (Fig 3). In contrast, no such relation was found in the reference population (Table 4).

**Table 4. Distribution of BMI categories, gestational weight gain categories, and birthweight z-scores in the reference and the pregestational diabetes mellitus population (160 and 49 participants, respectively).**

|  |  | Reference Median (range) | | | PGDM Median (range) | | |
|---|---|---|---|---|---|---|---|
| BMI (kg/m$^2$) |  | 23.0 (17.0–41.0) | | | 25.4 (19.8–44.1) | | |
| GWG/week (kg/week) |  | 0.37 (0.01–0.73) | | | 0.46 (-0.14–0.95) | | |
|  | Category | n | % | Mean BW z-score | n | % | Mean BW z-score |
| BMI | Underweight | 7 | 4.4 | -0.15 | 2 | 4.1 | 1.47 |
|  | Normal weight | 101 | 63.1 | -0.11 | 22 | 44.9 | 0.62 |
|  | Overweight | 43 | 26.9 | 0.17 | 14 | 28.6 | 2.02 |
|  | Obese | 9 | 5.6 | -0.52 | 11 | 22.4 | 0.59 |
| $p^*$ |  |  |  | 0.224 |  |  | 0.001* |
| GWG | Insufficient | 47 | 29.4 | -0.16 | 6 | 12.2 | 0.31 |
|  | Appropriate | 61 | 38.1 | -0.08 | 16 | 32.7 | 0.60 |
|  | Excessive | 47 | 29.4 | 0.10 | 27 | 55.1 | 1.48 |
| $p^*$ |  |  |  | 0.556 |  |  | 0.008* |
| Total group |  | 160 |  | -0.06 (-3.02–1.81) | 49 |  | 1.05 (-2.15–5.82) |

PGDM, pregestational diabetes; BMI, body mass index (kg/m$^2$); BMI categories defined by Institute Of Medicine guidelines: BMI; underweight (<18.5), normal weight (18.5–24.9), overweight (25–29.9), obese (≥30); BW, birthweight; GWG, weekly gestational weight gain; GWG category defined by Institute Of Medicine: insufficient, appropriate, excessive; z-score, standard deviation score

* $p < 0.05$, difference between BMI and GWG categories within the reference and the PGDM populations tested by ANOVA.

**BMI, gestational weight gain and fetal venous liver flow.** In the study population, pre-pregnancy BMI and gestational weight gain substantially modified fetal venous liver flow, compared with what was seen in the low-risk reference population (Fig 4, and Tables 5 and 6).

In the study population, the overweight group had the highest umbilical flow to liver, left portal vein blood velocity, and thus the highest total venous flow to liver, but the lowest relative portal contribution (Fig 4 and Table 5).

Further, in the study population, gestational weight gain was significantly associated with fetal venous flow. Women with excessive gestational weight gain had the highest umbilical flow, umbilical flow to liver, and left portal vein velocity, while the total venous flow to liver was highest in the appropriate weight gain group (Fig 4 and Table 6). Those with appropriate and excessive gestational weight gain had the highest umbilical flow to liver (Fig 4).

## Discussion

We found that in PGDM pregnancies, high birthweight was related to increased umbilical flow to the fetal liver. For similar volumes of umbilical flow to the liver, the association of flow with birthweight was stronger in PGDM pregnancies compared with the reference. Interestingly, with increasing BMI and gestational weight gain the umbilical flow to the liver increased, but at extreme BMI, (obesity), this relation seemed to break down as both flows (Fig 4) and birthweights were lower (Fig 3).

The results are in line with experimental studies showing that increased umbilical flow to the fetal liver, leads to increased insulin-like growth factor 1 and 2 production and a correspondingly augmented somatic growth of the fetus [10,15]. This concept is supported by human studies showing that the fetal liver, with its umbilical venous supply, plays a key role in fetal growth regulation and fat deposition, even in accelerated fetal growth of non-diabetic mothers [11,15,16]. In our study of PGDM pregnancies, these mechanisms were augmented and powerfully modified by maternal BMI and gestational weight gain.

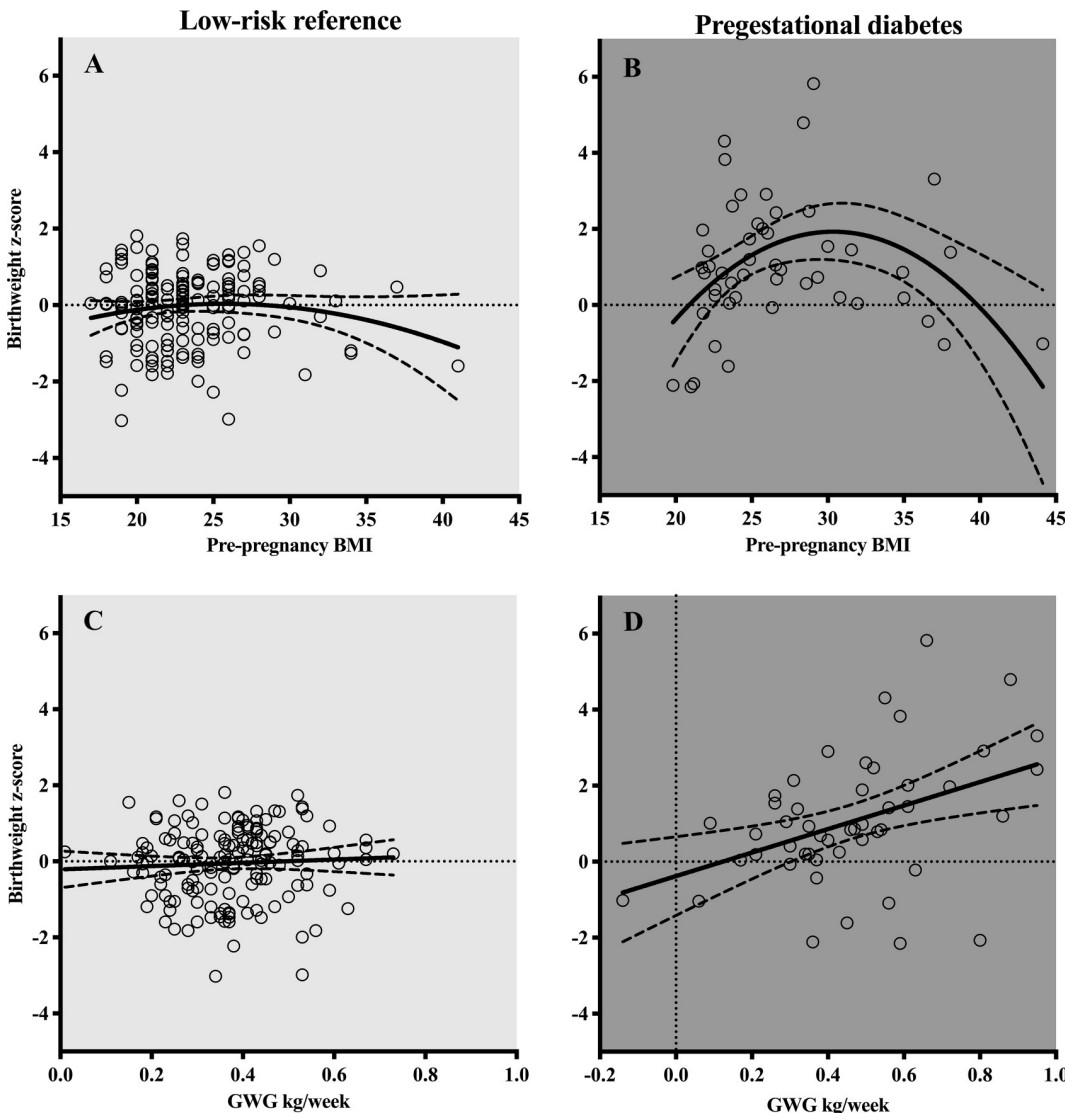

**Fig 3. Relation between body mass index and gestational weight gain and birthweight z-score in the reference and pregestational diabetes populations (160 and 49 participants, respectively).**

The present findings are also in agreement with the previously reported synergism between high BMI, excessive gestational weight gain and PGDM leading to increased risk of large for gestational age offspring [7,26]; here we have added to the understanding of the pathophysiology that these mechanisms seem, to a large extent to operate through the fetal venous liver circulation. Furthermore, the impact of gestational weight gain on birthweight is independent of glycemic control and BMI in women with PGDM [8,27]. This is in line with our study, where glycemic control (HbA$_{1C}$) did not differ between the BMI or gestational weight gain categories. Rather, it seemed to be through augmentation of umbilical flow to the liver that BMI and weight gain affected birthweight (Figs 2 and 4).

The level of glucose exposure influences fetal growth, via modulation of blood flow to the fetal liver [28]. In low-risk pregnancies, a maternal oral glucose load increased umbilical and venous liver flows and the response was associated with large fetal abdominal circumference [29]. The maternal metabolic status seems to influence the fetal response to a maternal meal:

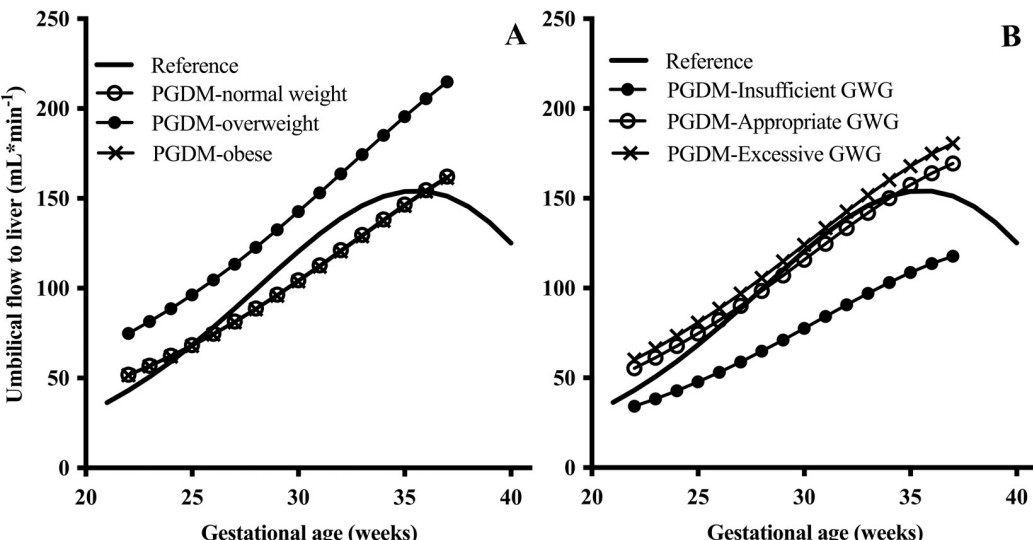

**Fig 4. Development of umbilical flow to the fetal liver and its association with BMI or gestational weight gain in pregnancies with pregestational diabetes mellitus (n = 49) compared with that of the reference pregnancies (n = 160).**

in a healthy population, increased umbilical flow to liver was observed in normal weight, but not in overweight mothers [17]. Further, the maternal-fetal glucose gradient correlated negatively with umbilical flow to liver in pregnancies of normal weight, but not overweight women [18]. Inadequate glycemic control is more frequent in patients with type 1 diabetes with high BMI [30]. Although HbA$_{1C}$ was not higher in those with overweight or excessive weight gain in our study population, episodes of hyperglycemia are more frequent in these groups [31] and this could be related to the observed increased umbilical venous flow and higher birthweights [29,32]. Further, defect epinephrine counter-regulation during hypoglycemia in PGDM pregnancies contributes to excessive fetal growth [33], probably through compensatory bouts of calorie intake with subsequent fetal hyperinsulinemia.

In women with PGDM, gestational weight gain contributes to excessive fetal growth, independent of maternal BMI and glycemic control [8,34]. The mechanisms are not completely understood, but additional nutrients delivery (fatty acids and amino acids) and altered leptine levels are suggested to contribute to accelerated fetal growth [35,36]. In addition to nutritional and hormonal influence, the present study suggests fetal blood flow as a possible link between maternal GWG and increased birth weight: Our PGDM population with excessive weight gain had higher umbilical flow to the liver and also higher birthweights (Figs 2 and 4). In PGDM pregnancies the augmented venous liver flow in the fetus seems to enhance fetal growth and fat deposition, possibly as a combined effect of increased flow and increased glucose and lipid content [15,37].

The association between low umbilical flow and growth perturbation is well documented [38]. In our study, obesity was not associated with augmented fetal growth, in contrast to the fetuses of overweight women (Table 4 and Fig 3). Although lower birthweights in the obese women could seem advantageous (since several perinatal risks in PGDM pregnancies are associated with macrosomia [2]), we believe that lower birthweights in those with PGDM *and* obesity more likely reflect relative placental insufficiency added to the adverse effects of fetal hyperglycemia. The finding also corroborates the disadvantage of inflammation commonly shown in obesity and linked to placental changes with adverse outcome [39]. A clinical

**Table 5. Fetal venous liver flow according to pre-pregnancy BMI categories in the reference and pregestational diabetes mellitus populations (160 and 49 participants, respectively).**

| | | Flow z-score | | | | | |
|---|---|---|---|---|---|---|---|
| | BMI category | Reference population | | | Pregestational diabetes population | | |
| | | n | mean | CI | n | mean | CI |
| Umbilical flow | normal | 363 | 0.016 | -0.09–0.12 | 91 | 0.228 | -0.09–0.55 |
| | overweight | 155 | -0.121 | -0.28–0.04 | 58 | 0.703 | 0.30–1.11 |
| | obese | 30 | 0.327 | -0.04–0.69 | 43 | 0.206 | -0.26–0.67 |
| *p* | | | *0.130* | | | *0.144* | |
| Umbilical flow to liver | normal | 353 | 0.028 | -0.08–0.13 | 59 | 0.101 | -0.31–0.52 |
| | overweight | 150 | -0.143 | -0.30–0.02 | 40 | 0.906 | 0.40–1.41 |
| | obese | 26 | 0.218 | -0.17–0.61 | 24 | 0.063 | -0.59–0.71 |
| *p* | | | *0.190* | | | *0.033** | |
| Ductus venosus flow | normal | 344 | 0.132 | -0.26–0.52 | 70 | -0.145 | -0.59–0.30 |
| | overweight | 147 | 0.092 | -0.01–0.20 | 42 | -0.234 | -0.81–0.34 |
| | obese | 27 | -0.171 | -0.33 - -0.01 | 26 | -0.870 | -1.60 - -0.14 |
| *p* | | | *0.005** | | | *0.237* | |
| Ductus venosus flow fraction | normal | 340 | -0.072 | -0.52–0.28 | 59 | -0.158 | -0.56–0.24 |
| | overweight | 143 | 0.207 | -0.18–0.04 | 40 | -0.759 | -1.25 - -0.27 |
| | obese | 26 | 0.088 | -0.30–0.48 | 24 | -0.550 | -1.18–0.08 |
| *p* | | | *0.042** | | | *0.160* | |
| Left portal vein flow velocity | normal | 349 | -0.178 | -0.57–0.21 | 100 | 0.591 | 0.35–0.84 |
| | overweight | 149 | 0.002 | -0.10–0.11 | 54 | 0.844 | 0.51–1.18 |
| | obese | 30 | 0.037 | -0.12–0.20 | 48 | 0.507 | 0.15–0.86 |
| *p* | | | *0.801* | | | *0.343* | |
| Portal vein flow | normal | 354 | -0.014 | -0.12–0.09 | 51 | 0.315 | -0.25–0.88 |
| | overweight | 149 | 0.062 | -0.10–0.22 | 30 | .0385 | -0.35–1.12 |
| | obese | 30 | 0.195 | -0.17–0.56 | 13 | -0.157 | -1.27–0.96 |
| *p* | | | *0.660* | | | *0.706* | |
| Portal vein fraction | normal | 332 | -0.043 | -0.15–0.07 | 42 | 0.066 | -0.80–0.61 |
| | overweight | 140 | 0.163 | -0.01–0.33 | 25 | -0.446 | -1.15–0.26 |
| | obese | 26 | -0.090 | -0.48–0.30 | 9 | 0.102 | -1.08–1.28 |
| *p* | | | *0.159* | | | *0.491* | |
| Total venous flow to liver | normal | 333 | 0.012 | -0.10–0.12 | 42 | 0.304 | -0.18–0.78 |
| | overweight | 142 | -0.115 | -0.28–0.05 | 25 | 1.087 | 0.47–1.71 |
| | obese | 26 | 0.238 | -0.15–0.63 | 9 | -0.160 | -1.19–0.88 |
| *P* | | | *0.213* | | | *0.061* | |

n, total number of observations in reference (n = 160) and study population (49)

* *p*-value <0.05, Fetal flow z-score according to body mass index (BMI) categories within each population tested by ANOVA; n, number of observations; Flow (mL/min); Flow velocity, time-averaged maximum velocity (cm/sec); BMI categorized as: normal (18.5–24.9), overweight (25–29.9) or obese (≥30) (underweight BMI category was excluded).

UV flow to liver = UV flow- DV flow.

Total venous flow to liver = UV flow to liver + PV flow.

Ductus venosus flow fraction = DV flow/UV flow*100.

Portal vein fraction = PV flow/ Total venous liver flow*100.

message emanates from these results; absence of macrosomia in PGDM pregnancies of obese women, does not exclude perinatal risks but calls for continued attentiveness [39].

In low-risk populations, low maternal BMI, low weight gain and low maternal skinfold thickness were associated with a compensating increase in umbilical flow to liver near term

**Table 6. Fetal venous liver flow according to gestational weight gain categories in the reference and pregestational diabetes mellitus populations (160 and 49 participants, respectively).**

| | | Flow z-score | | | | | |
|---|---|---|---|---|---|---|---|
| | GWG category | Reference population | | | Pregestational diabetes population | | |
| | | n | Mean | CI | n | Mean | CI |
| Umbilical flow | insufficient | 172 | -0.078 | -0.23–0.08 | 19 | -0.613 | -1.31–0.08 |
| | appropriate | 218 | 0.071 | -0.07–0.21 | 63 | 0.440 | 0.06–0.82 |
| | excessive | 164 | 0.017 | -0.14–0.08 | 110 | 0.494 | 0.21–0.78 |
| *p* | | | *0.364* | | | *0.015* | |
| Umbilical flow to liver | insufficient | 171 | -0.068 | -0.22–0.09 | 14 | -1.087 | -1.91–0.26 |
| | appropriate | 209 | 0.075 | -0.06–0.21 | 40 | 0.425 | -0.06–0.92 |
| | excessive | 157 | -0.025 | -0.18–0.13 | 69 | 0.608 | 0.24–0.98 |
| *p* | | | *0.367* | | | *0.001* | |
| Ductus venosus flow | insufficient | 172 | 0.071 | -0.08–0.22 | 17 | -0.437 | -1.34–0.47 |
| | appropriate | 205 | 0.031 | -0.11–0.17 | 45 | -0.556 | -1.11–0.01 |
| | excessive | 151 | -0.114 | -0.28–0.47 | 76 | -0.133 | -0.56–0.30 |
| *p* | | | *0.229* | | | *0.474* | |
| Ductus venosus fraction | insufficient | 171 | -0.009 | -0.16–0.14 | 14 | 0.272 | -0.55–1.10 |
| | appropriate | 200 | 0.061 | -0.20–0.08 | 40 | -0.583 | -1.07 - -0.09 |
| | excessive | 148 | 0.091 | -0.07–0.26 | 69 | -0.484 | -0.86 - -0.11 |
| *p* | | | *0.381* | | | *0.196* | |
| Left portal vein flow velocity | insufficient | 170 | -0.083 | -0.24–0.07 | 24 | 0.683 | 0.19–1.18 |
| | appropriate | 210 | 0.053 | -0.08–0.19 | 66 | 0.287 | -0.01–0.59 |
| | excessive | 153 | 0.057 | -0.10–0.22 | 112 | 0.836 | 0.61–1.07 |
| *p* | | | *0.340* | | | *0.017* | |
| Portal vein flow | insufficient | 173 | -0.110 | -0.26–0.04 | 12 | 0.030 | -1.13–1.12 |
| | appropriate | 208 | 0.050 | -0.09–0.19 | 31 | 0.587 | -0.13–1.31 |
| | excessive | 158 | 0.132 | -0.02–0.29 | 51 | 0.138 | -0.42–0.70 |
| *p* | | | *0.077* | | | *0.564* | |
| Portal vein fraction | insufficient | 170 | -0.037 | -0.19–0.12 | 10 | 1.002 | -0.08–2.01 |
| | appropriate | 193 | -0.046 | -0.19–0.10 | 24 | 0.083 | -0.62–0.78 |
| | excessive | 144 | 0.111 | -0.06–0.28 | 42 | -0.463 | -0.99–0.07 |
| *p* | | | *0.315* | | | *0.050* | |
| Total venous flow to liver | insufficient | 171 | -0.066 | -0.22–0.09 | 10 | -0.887 | -1.84–0.07 |
| | appropriate | 193 | 0.053 | -0.09–0.20 | 24 | 0.917 | 0.30–1.53 |
| | excessive | 146 | 0.012 | -0.15–0.18 | 42 | 0.604 | 0.14–1.07 |
| *p* | | | *0.536* | | | 0.008 | |

Fetal flow z-scores according to weekly gestational weight gain (GWG) categories within each population tested by ANOVA; n, total number of observations; Flow (mL/min); Flow velocity, time-averaged maximum velocity (cm/sec); Gestational weight gain (GWG) categories defined by the institute of medicine: insufficient, appropriate or excessive.

UV flow to liver = UV flow- DV flow.

Total venous flow to liver = UV flow to liver + PV flow.

Ductus venosus flow fraction = DV flow/UV flow*100.

Portal vein fraction = PV flow/ Total venous liver flow*100.

[14,16,40]. Such prioritization, in situations of restricted maternal nutritional supply, is thought to be a protective mechanism to enhance the offspring fat accretion [16,37]. In PGDM pregnancies however, such increase in umbilical flow to liver in combination with the hyperglycemic in-utero-metabolic environment, augments the fetal fat deposition [16].

The risks of metabolic syndrome, obesity and diabetes in individuals born from PGDM pregnancies, are not explained by genetic dispositions alone [41–43]. Important additional determinants are found in the in-utero metabolic programming that conditions health risks in postnatal life, increasingly supported by emerging epigenetic studies in the offspring of women with diabetes in pregnancy [44,45]. In this scenario, the fetal liver circulation stands out as an example of adaptive mechanisms in the interphase between umbilical blood flow and endocrine liver function, and metabolism sensitive to environmental cues, with possible consequences for child development and future health [32,46,47].

The strengths of this study are the unselected populations of low-risk (reference) and PGDM pregnancies [14], the identical and validated ultrasound and Doppler methods applied to both populations and the prospective longitudinal design.

Self-reported pre-pregnancy weight could introduce a recall bias but is widely used in research and allows comparison with other studies [8,48]. We consider the difference between the self-reported and measured weights at inclusion in the PGDM population (about 2 kilograms) to be plausible [49,50]. High BMI in our PGDM population hampered the ultrasound examination and reduced the success rate for the fetal flow measurements. A higher success rate in the leaner PGDM women may have skewed the study population towards normality, but this selection would reduce rather than increase the observed differences between the study- and reference populations. There were no differences in HbA$_{1C}$ between the group with missing and complete data, which makes selection bias by glycemic control less likely. In large population-based studies, pre-pregnancy BMI and gestational weight gain are associated with the risk of large for gestational age infants [51]. The absence of this association in our reference population might be due to the fact that the size of the association is too small for this sample size, or selection bias as the inclusions were healthy women, not random selection of the general population (Table 4). A possible limitation is that the study of the reference population was carried out almost ten years prior to the present study. Seven women in the study population used anti-hypertensive drugs which may influence maternal and feto-placental hemodynamics [52,53]. We considered the size of the study population too small for subgroup analyses of maternal ethnicity, the use of antihypertensive drugs or sex of the neonate.

In summary, increased umbilical flow to liver seems to be in the causal pathway to larger birthweights in PGDM pregnancies, and maternal overweight and excessive gestational weight gain augment this association. In obese women with PGDM however, birthweights in the normal range do not exclude perinatal risks as they are probably due to relatively blunted placental and metabolic resources.

## Acknowledgments

We acknowledge bioengineer Carol Cook for her practical assistance in the study. We thank the women that participated in the study. The department of Obstetrics and Gynecology, Haukeland University Hospital provided facilities and equipment to conduct this research.

## Author Contributions

**Conceptualization:** Agnethe Lund, Cathrine Ebbing, Svein Rasmussen, Torvid Kiserud, Jörg Kessler.

**Data curation:** Agnethe Lund.

**Formal analysis:** Agnethe Lund, Svein Rasmussen, Jörg Kessler.

**Funding acquisition:** Agnethe Lund, Cathrine Ebbing, Svein Rasmussen, Jörg Kessler.

**Investigation:** Agnethe Lund, Cathrine Ebbing, Elisabeth Qvigstad, Jörg Kessler.

**Methodology:** Agnethe Lund, Cathrine Ebbing, Svein Rasmussen, Jörg Kessler.

**Project administration:** Agnethe Lund, Jörg Kessler.

**Supervision:** Cathrine Ebbing, Svein Rasmussen, Elisabeth Qvigstad, Torvid Kiserud, Jörg Kessler.

**Writing – original draft:** Agnethe Lund, Cathrine Ebbing, Torvid Kiserud, Jörg Kessler.

**Writing – review & editing:** Agnethe Lund, Cathrine Ebbing, Svein Rasmussen, Elisabeth Qvigstad, Torvid Kiserud, Jörg Kessler.

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
