## [Decision Letter · Decision Letter 0]

19 May 2021

PONE-D-21-10502

Pre-gestational diabetes: maternal body mass index and gestational weight gain augments umbilical venous flow, fetal liver perfusion, and thus birthweight

PLOS ONE

Dear Dr. Ebbing,

Thank you for submitting your manuscript to PLOS ONE. After careful consideration, we feel that it has merit but does not fully meet PLOS ONE’s publication criteria as it currently stands. Therefore, we invite you to submit a revised version of the manuscript that addresses the points raised during the review process.

Please take into account all the remarks made by Reviewer 2 in your revised manuscript.

In addition, we note that the manuscript text contains numerous causal inferences, regarding both the hypothesis and the studied potential mechanisms. Although the study includes a comparison with a reference group, its observational design does not allow firm causative conclusions in our opinion, despite the likelihood and potential interest of the hypotheses. We would suggest that any terms like "effect", "impact", and other explicit causative wording be avoided and replaced by "associations". The potential mechanistic sequence described between the measured blood flows parameters and the birth weight should thus still be presented as an - eventually strong -  hypothesis. These changes in the wording should be applied to the entire text, including the manuscript title and the abstract.

We look forward to receiving your revised manuscript.

Kind regards,

Umberto Simeoni

Academic Editor

PLOS ONE

Journal Requirements:

3. Please provide additional details regarding participant consent. In the ethics statement in the Methods and online submission information, please ensure that you have specified whether consent was informed.

Reviewers' comments:

Reviewer's Responses to Questions

**Comments to the Author**

1. Is the manuscript technically sound, and do the data support the conclusions?

Reviewer #1: Yes

Reviewer #2: Yes

2. Has the statistical analysis been performed appropriately and rigorously? 

Reviewer #1: Yes

Reviewer #2: Yes

3. Have the authors made all data underlying the findings in their manuscript fully available?

Reviewer #1: Yes

Reviewer #2: Yes

4. Is the manuscript presented in an intelligible fashion and written in standard English?

Reviewer #1: Yes

Reviewer #2: Yes

5. Review Comments to the Author

Reviewer #1: The Authors in this prospective longitudinal observational study acess ,the modifying effects of maternal BMI and gestational weight gain ,on the venous perfusion of the fetal liver and birthweight. In this work they investigate fetal hemodynamics of liver perfusion evaluating the development of ductus venous, umbilical and portal blood flows during 2nd trimester of pregnancies with PGDM.

The results are very original and sound, explaning larger birthweights of fetuses in those pregnancies as part of perinatal programming of future cardiovascular risk of newborns of those pregnancies.

Reviewer #2: In their longituonal observational study in women with pregestational DM the authors identify umbilical venous bloodflow to be significantly associated with birthweight. Importantly, this association is augmented in women with PGDM when compared to appropriately selected controls.

The vast majority (90%) of women in the PGDM have T1DM with on average fairly good metabolic control (HbA1c 6.7%).

The authors report that the effects of maternal weight and weight gain on birthweight are associated with umbilical blood flow but not metabolic control, as indicated by HbA1c. It would be interesting to know, whether metabolic control per se impacts umbilical blood flow.

Although still considered the most important surrogate of glycemic control, HbA1c has several limitations in pregnancy. Can the authors present data or at least speculate whether glycemia per se, i.e. ambient glucose levels and/or glycemic variability affect umbilical venous blood flow and whether this could have affected their results.

The authors discuss a possible role of maternal prandial status on umbilical blood flow. Was this considered when the experiments were performed ?

The authors speculate on a possible association of hypoglycemic episodes, gestational weight gain, altered fuel supply, i.e. increased lipid/FFA delivery to the fetus. However, repeat hypoglycemia has been associated with fetal growth restriction. Please discuss.

The authors report that several patients suffered from chronic hypertension. Could the maternal intake of antihypertensives, presumably mostly betablockers, have influenced the results of the study ?

6. PLOS authors have the option to publish the peer review history of their article (what does this mean?). If published, this will include your full peer review and any attached files.

Reviewer #1: No

Reviewer #2: No

---

## [Author Response · Author response to Decision Letter 0]

28 Jun 2021

Dear Academic Editor Umberto Simeoni.

We thank for the review of our paper (PONE-D-21-10502)

"Pre-gestational diabetes: maternal body mass index and gestational weight gain augments umbilical venous flow, fetal liver perfusion, and thus birthweigh). We appreciate the opportunity to improve our manuscript and to reply to comments.

We have discussed each point raised and carried out changes in the manuscript. Please find our responses below.

“Dear Dr. Ebbing,

Thank you for submitting your manuscript to PLOS ONE. After careful consideration, we feel that it has merit but does not fully meet PLOS ONE’s publication criteria as it currently stands. Therefore, we invite you to submit a revised version of the manuscript that addresses the points raised during the review process. 

Please take into account all the remarks made by Reviewer 2 in your revised manuscript.

Response: Thank you. We will explain how the remarks has been accounted for in the revised manuscript.

In addition, we note that the manuscript text contains numerous causal inferences, regarding both the hypothesis and the studied potential mechanisms. Although the study includes a comparison with a reference group, its observational design does not allow firm causative conclusions in our opinion, despite the likelihood and potential interest of the hypotheses. We would suggest that any terms like "effect", "impact", and other explicit causative wording be avoided and replaced by "associations". The potential mechanistic sequence described between the measured blood flows parameters and the birth weight should thus still be presented as an - eventually strong - hypothesis. These changes in the wording should be applied to the entire text, including the manuscript title and the abstract.

Reply: Thank you! We carried out changes throughout the manuscript, and these are marked in the new version of the manuscript with Track Changes. The changes implied a revised title and conclusion in the abstract.

Title: “Pre-gestational diabetes: maternal body mass index and gestational weight gain are associated with augmented umbilical venous flow, fetal liver perfusion, and thus birthweight.”

Revised conclusions abstract:” The results support the hypothesis that umbilical flow to the fetal liver is a key determinant for fetal growth and birthweight modifiable by maternal factors. Maternal pre-gestational diabetes mellitus seems to augment this influence as shown with body mass index and gestational weight gain.”

Reply: We have made sure that our manuscript meets PLOS one style requirements.

 Response: We have identified a typo and added some references to the list in line with the changes we have carried out in response to the reviewer's comments.

3. Please provide additional details regarding participant consent. In the ethics statement in the Methods and online submission information, please ensure that you have specified whether consent was informed.

Reply: We have specified in the Methods section that participants had informed consent. 

Reply: Please see the revised cover letter. 

Reviewers' comments:

Reviewer's Responses to Questions

Comments to the Author

1. Is the manuscript technically sound, and do the data support the conclusions?

Reviewer #1: Yes

Reviewer #2: Yes

2. Has the statistical analysis been performed appropriately and rigorously? 

Reviewer #1: Yes

Reviewer #2: Yes

3. Have the authors made all data underlying the findings in their manuscript fully available?

Reviewer #1: Yes

Reviewer #2: Yes

4. Is the manuscript presented in an intelligible fashion and written in standard English?

Reviewer #1: Yes

Reviewer #2: Yes

5. Review Comments to the Author

Reviewer #1: The Authors in this prospective longitudinal observational study acess ,the modifying effects of maternal BMI and gestational weight gain ,on the venous perfusion of the fetal liver and birthweight. In this work they investigate fetal hemodynamics of liver perfusion evaluating the development of ductus venous, umbilical and portal blood flows during 2nd trimester of pregnancies with PGDM.

 The results are very original and sound, explaning larger birthweights of fetuses in those pregnancies as part of perinatal programming of future cardiovascular risk of newborns of those pregnancies.

Reviewer #2: In their longituonal observational study in women with pregestational DM the authors identify umbilical venous bloodflow to be significantly associated with birthweight. Importantly, this association is augmented in women with PGDM when compared to appropriately selected controls.

 The vast majority (90%) of women in the PGDM have T1DM with on average fairly good metabolic control (HbA1c 6.7%). 

The authors report that the effects of maternal weight and weight gain on birthweight are associated with umbilical blood flow but not metabolic control, as indicated by HbA1c. It would be interesting to know, whether metabolic control per se impacts umbilical blood flow.

 Although still considered the most important surrogate of glycemic control, HbA1c has several limitations in pregnancy. Can the authors present data or at least speculate whether glycemia per se, i.e. ambient glucose levels and/or glycemic variability affect umbilical venous blood flow and whether this could have affected their results.

Reply: Thank you for this comment. Unfortunately, we do not have access to ambient glucose levels or variability in our study population at the time of ultrasound and Doppler evaluation of the UV flow. We have explored the association between maternal glycemic control and fetal venous liver flow in two previous publications (Lund A et al Acta Obstet Gynecol 2018 Aug;97(8):1032-1040; Lund A. et al, PloS One 2019; 14(3), e0211788). We found that there was a negative relation between maternal HbA1c and the ductus venosus flow velocity, flow volume and shunt fraction, especially near term. Further, maternal HbA1c had a positive association to left portal vein flow velocities and a negative relation to the contribution of portal blood flow to the venous liver flow. Fetuses of women with high HbA1c had higher umbilical blood flow, this association was however not statistically significant (Fig. 1). We may speculate that in women with HbA1c levels that are low, shunt fraction and UV flow resembles the distribution in women without diabetes.

Fig. 1: Umbilical blood flow according to maternal 1st trimester HbA1c in PGDM 

The authors discuss a possible role of maternal prandial status on umbilical blood flow. Was this considered when the experiments were performed ?

Reply: Unfortunately, we did not register the timing or composition of the last meal in relation to the Doppler/ultrasound evaluation. We realize in hindsight that this would have been an interesting aspect. The women did not seldomly check their sugar levels during examination, and some even had a snack on the examination bench during examination when glucose levels were low.

The authors speculate on a possible association of hypoglycemic episodes, gestational weight gain, altered fuel supply, i.e. increased lipid/FFA delivery to the fetus. However, repeat hypoglycemia has been associated with fetal growth restriction. Please discuss.

Reply: Thank you for this important comment. We agree, excessive gestational weight gain is independently associated with increased risk of large for gestational age neonates, and hypoglycemia is associated with low birthweight and placental weights. Excessive GWG is associated with hypertension. Altered fuel supply to the fetus as a response to hypoglycemia may be influenced by maternal factors such as duration of the disease, responder status and complications (nephropathy, retinopathy). There is a clinical impression that frequent hypoglycemic events lead to extra carbohydrate intake which again may add to increased gestational weight gain. The present study does not contain data on nutritional factors (lipids and fasting glucose levels) and cannot answer the question whether episodes of hypoglycemia alter the substrate supply to the fetus, but this should be explored. We have now elaborated on this and adjusted the discussion.

 “ Further, defect epinephrine counter-regulation during hypoglycemia in PGDM pregnancies contributes to excessive fetal growth [33], probably through compensatory bouts of calorie intake with subsequent fetal hyperinsulinemia. 

In women with PGDM, gestational weight gain contributes to excessive fetal growth, independent of maternal BMI and glycemic control [8, 34]. The mechanisms are not completely understood, but additional nutrients delivery (fatty acids and amino acids) and altered leptine levels are suggested to contribute to accelerated fetal growth [35,36]. In addition to nutritional and hormonal influence, the present study suggests fetal blood flow as a possible link between maternal GWG and increased birth weight”

The authors report that several patients suffered from chronic hypertension. Could the maternal intake of antihypertensives, presumably mostly betablockers, have influenced the results of the study ?

Reply: Absolutely, we agree that we cannot rule out that the use of antihypertensive medicaments may influence UV flow. In our study population seven of the participants had chronic hypertension. We did not perform any sub-analyses in this group due to the small size of the population. We acknowledge that maternal use of antihypertensives may alter fetal distribution of umbilical flow, however, human evidence is scarce, and evidence in pregnancies with DM1 are lacking. Jouppila,P. et al have performed studies on antihypertensives and the effect on fetal and maternal hemodynamics. Their findings in women and fetuses with preeclampsia suggested that labetalol reduces maternal blood pressure without interfering with the placental or fetal blood flow (Jouppila P et al Br J Obstet Gynecol 1986 Jun;93(6):543-7). While evidence from experimental studies in sheep showed that labetalol reduced placental volume flow, and increased placental vascular resistance. (Erkinaro T et al Reprod Sci 2009 Aug;16(8):749-57). We have now inserted a sentence in the discussion section about this issue.

“...study. Seven women in the study population used anti-hypertensive drugs which may influence maternal and feto-placental hemodynamics [52, 53]. We considered the size of the study population too small for subgroup analyses of maternal ethnicity, the use of antihypertensive drugs or sex of the neonate.”

6. PLOS authors have the option to publish the peer review history of their article (what does this mean?). If published, this will include your full peer review and any attached files.

Do you want your identity to be public for this peer review? For information about this choice, including consent withdrawal, please see our Privacy Policy.

Reviewer #1: No

Reviewer #2: No

Response: All figure files were approved in PACE

---

## [Decision Letter · Decision Letter 1]

2 Aug 2021

Pre-gestational diabetes: maternal body mass index and gestational weight gain are associated with augmented umbilical venous flow, fetal liver perfusion, and thus birthweight.

PONE-D-21-10502R1

Dear Dr. Ebbing,

We’re pleased to inform you that your manuscript has been judged scientifically suitable for publication and will be formally accepted for publication once it meets all outstanding technical requirements.

Kind regards,

Umberto Simeoni

Academic Editor

PLOS ONE

Additional Editor Comments (optional):

Reviewers' comments:

Reviewer's Responses to Questions

**Comments to the Author**

1. If the authors have adequately addressed your comments raised in a previous round of review and you feel that this manuscript is now acceptable for publication, you may indicate that here to bypass the “Comments to the Author” section, enter your conflict of interest statement in the “Confidential to Editor” section, and submit your "Accept" recommendation.

Reviewer #2: All comments have been addressed

2. Is the manuscript technically sound, and do the data support the conclusions?

Reviewer #2: Yes

3. Has the statistical analysis been performed appropriately and rigorously? 

Reviewer #2: Yes

4. Have the authors made all data underlying the findings in their manuscript fully available?

Reviewer #2: Yes

5. Is the manuscript presented in an intelligible fashion and written in standard English?

Reviewer #2: Yes

6. Review Comments to the Author

Reviewer #2: The comments and suggestions have been properly and thoroughly adressed. According changes have been made to the manuscript.

7. PLOS authors have the option to publish the peer review history of their article (what does this mean?). If published, this will include your full peer review and any attached files.

Reviewer #2: No

---

## [Editor Report · Acceptance letter]

6 Aug 2021

PONE-D-21-10502R1 

Pre-gestational diabetes: maternal body mass index and gestational weight gain are associated with augmented umbilical venous flow, fetal liver perfusion, and thus birthweight. 

Dear Dr. Ebbing:

I'm pleased to inform you that your manuscript has been deemed suitable for publication in PLOS ONE. Congratulations! Your manuscript is now with our production department. 

Kind regards, 

on behalf of

Dr. Umberto Simeoni 

Academic Editor

PLOS ONE